# Variational Autoencoder-Based Model Predictive Control for Automated Fluid Resuscitation

Elham Estiri
*College of Aeronautics and Engineering*
*Kent State University*
Kent, OH, USA
eestiri@kent.edu

Hossein Mirinejad
*College of Aeronautics and Engineering*
*Kent State University*
Kent, OH, USA
hmiri@kent.edu

*Abstract*—This paper presents a novel approach for automated fluid resuscitation by modeling hemodynamics with a machine learning method and controlling it with a model predictive control (MPC) algorithm. The modeling framework, called the robust nonlinear state-space modeling (RNSSM), uses variational autoencoders to predict hemodynamic responses from limited and noisy critical care data during hemorrhage resuscitation. The MPC controller, designed for the RNSSM models, leverages its predictive capabilities for precise control of fluid dosages in resuscitation. Simulation results demonstrate the potential of this approach in improving the safety and efficacy of fluid resuscitation in critical care settings.

*Index Terms*—Fluid resuscitation, robust nonlinear state-space modeling, autoencoder learning, variational autoencoder, model predictive control

## I. INTRODUCTION

Fluid resuscitation is a therapeutic approach used to restore the volume of fluid in the circulatory system for critical care patients [1]. Precise control of fluid dosages is crucial for ensuring patient safety and promoting recovery [2]. Computational modeling and advanced control methods can aid in regulating the volume and rate of fluid delivery. These techniques help reduce delays in care, minimize dosing errors, and decrease the cognitive load on clinicians, resulting in improved patient safety and outcomes.

In recent years, there has been growing interest in applying advanced mathematical models and controllers to fluid resuscitation [2]–[10]. Most methods suggested for automated fluid resuscitation have been control-oriented low-order models. These models are limited in handling complex, nonlinear physiological processes, lack adaptability to individual patient variations, and struggle with real-time data integration. In contrast, machine learning-based models potentially offer superior predictive performance, scalability, and real-time adaptability for complex physiological systems such as fluid resuscitation. However, there has been very limited work on applying machine learning to fluid therapy [5]. The suggested machine learning methods often leverage model-free reinforcement learning (RL) [5], which requires a substantial amount of

This material is based upon work supported by the National Science Foundation CAREER Award under Grant No. 2340139 and by the National Science Foundation Engineering Research Initiation (ERI) Award under Grant No. 2138929.

clinical data and relies on the unrealistic assumption of access to a patient simulator as an environment for interacting with the RL controller [5]–[7].

Motivated by the above discussions, we introduce a pioneering approach to fluid resuscitation by developing a novel machine learning-based model for hemodynamic identification and a model predictive control (MPC) algorithm for fluid dosage adjustment. The modeling framework, called robust nonlinear state-space modeling (RNSSM), seamlessly integrates autoencoder learning and variational Gaussian inference (VGI) within a variational autoencoder (VAE) platform to develop nonlinear state-space models from limited and noisy critical care data. The proposed VAE framework is highly amenable to closed-loop control design, utilizing an MPC algorithm for optimal prediction of hemodynamic responses to fluid changes while accounting for dosing and model constraints.

The MPC's ability to incorporate constraints into the control problem ensures that the control actions are not only optimal but also safe and feasible in real-world scenarios. For instance, desired ranges for the control inputs (drug dosage) and the system outputs (physiological responses) can be specified as constraints within the MPC formulation. Additionally, the VAE's capability to capture complex, nonlinear relationships within physiological systems enhances predictive accuracy, allowing for more precise and anticipatory control of fluid delivery. Furthermore, the proposed VAE-based MPC is data-efficient, requiring less data to build an effective model compared to model-free RL methods, while offering greater stability and robustness than low-order lumped-parameter models. To the best of the authors' knowledge, this is the first attempt at developing VAE-based MPC in fluid resuscitation.

## II. METHODOLOGY

In this section, we present our proposed methodology for automatically regulating mean arterial pressure (MAP) responses to fluid infusion in hemorrhage scenarios using RNSSM modeling framework and MPC algorithm.

### A. Robust Nonlinear State-Space Modeling (RNSSM)

We propose a novel modeling methodology to predict robust, subject-specific MAP responses to fluid infusion. The

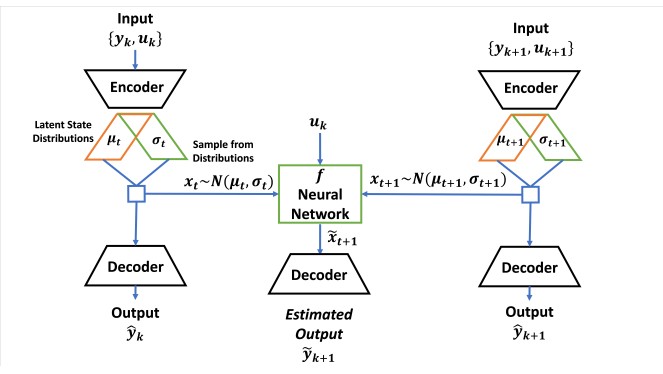

Fig. 1. The RNSSM framework integrates autoencoder learning and variational Gaussian inference to identify hemodynamics from limited and noisy clinical data.

RNSSM methodology is centered on identifying reliable nonlinear state-space models from limited, noisy clinical data using machine learning algorithms. The model is a multiple-input/multiple-output nonlinear state-space model of the form:

$$x_{k+1} = f(x_k, u_k, \theta) + v_k$$
$$y_k = g(x_k, u_k, \theta) + w_k \tag{1}$$

where $x_k$ is the (hidden) state variable, $u_k$ denotes the observed input, and $y_k$ represents the measured output. The functions $f(.)$ and $g(.)$ capture the state transition and output measurement, respectively, while $\theta$ represents a vector of unknown parameters. The terms $v_k$ and $w_k$ account for disturbance and measurement noise, respectively, both described by Gaussian probability density functions.

To properly capture complex hemodynamic relationships, the RNSSM approach integrates autoencoder learning with VGI techniques to identify nonlinear state-space models. These models will be used in designing the MPC algorithm in the next step, as will be shown later.

The autoencoder is an artificial neural network (ANN) that compresses input data into a lower-dimensional representation (encoding) and then reconstructs it to its original form (decoding). It is commonly used for tasks such as dimensionality reduction, feature extraction, and data denoising. However, regular autoencoders struggle to handle uncertainties from external sources (e.g., measurement noise) and internal sources (e.g., unmodeled dynamics), which is a major issue in noise-distorted clinical data. To address this, we integrated VGI techniques into the autoencoder framework, resulting in variational autoencoders (VAEs). A VAE is highly effective in learning a probabilistic distribution of the dataset in the latent space, allowing for better control over training data [11].

The RNSSM model consists of three main components, as shown in Fig. 1. First, there is a multi-layer ANN encoder that predicts $x_k$ from $I_{k-1}$, where $I_{k-1}$ is the input sequence including $\{y_{k-1}, u_{k-1}\}$. Second, a multi-layer ANN decoder is used for predicting $y_k$ from $x_k$. Finally, a bridge network, which is also a multi-layer ANN model, is responsible for modeling the function $f$ that maps $x_k$ to $x_{k+1}$.

In the proposed framework, the focus is on approximating the true posterior distribution $p_\theta(x|y)$, where $x$ denotes latent variables and $y$ is the observed data. Computing the true posterior $p_\theta(x|y)$ is analytically intractable, prompting the introduction of a variational inference that approximates the posterior using a simpler variational distribution $q_\phi(x|y)$, parameterized by $\phi = (\mu, \sigma)$. Here, $\mu$ and $\sigma$ denote the mean and standard deviation of the distribution, respectively. Their values are typically set to establish the prior distribution as a standard normal distribution, i.e., $\mu = 0$ and $\sigma = 1$. The training goal for a VAE is to determine model parameters making the variational distribution $q_\phi(x|y)$ closely match the true posterior distribution $p_\theta(x|y)$. This is achieved by minimizing the Kullback-Leibler (KL) divergence between the two distributions [12].

### B. Model Predictive Control Algorithm

We developed an MPC algorithm to control hemodynamic responses during fluid resuscitation. After identifying a model in the state-space form, several state feedback control techniques can be utilized to control the output variable. Among these techniques, MPC stands out for its flexibility and suitability in handling multivariable systems with constraints on input and output variables [13]. Additionally, MPC allows for the incorporation of predictive models to anticipate future states, enabling more precise and adaptive management of patient hemodynamics in real-time. This capability is particularly crucial in clinical settings where patient conditions can change rapidly and unpredictably.

To handle nonlinear systems, we employ the Quasi-Linear Parameter-Varying (quasi-LPV) structure [14]. This structure allows us to compute the system matrices directly using the nominal values of the state variables and inputs at each step of the prediction horizon. This approach enhances the accuracy of capturing the system's dynamic variations, thereby improving the performance of the MPC. The quasi-LPV model is unique in that the system matrices (A, B, C) are functions of the system states and inputs. This feature enables the model to accurately represent the system's dynamic variations [15]. Within the context of this framework, (1) can be represented as follows:

$$x_{k+1} = A(x_k, u_k)[x'_k, 1]' + B(x_k, u_k)u_k$$
$$y_k = C(x_k, u_k)[x'_k, 1]' \tag{2}$$

This representation provides a more accurate and flexible framework for control design, particularly in the context of MPC. The MPC controller used in this work begins by taking the following: prediction horizon $n_p$, control horizon $n_m$, weight metrics $W_y$ for the output, $W_u$ for the input, and $W_{\Delta u}$ for the change of input, the output reference signal $r_t$, and the current state estimate $x_t$. The algorithm first computes the sequence of predicted states $\{\hat{x}_{t+1}, ..., \hat{x}_{t+n_p}\}$ given the current guess of the input sequence $\{\hat{u}_t, ..., \hat{u}_{t+n_p-1}\}$, with $\hat{u}_{t+k} = \hat{u}_{t+n_m-1}$ for all $k, ..., n_m - 1$. Then, it computes the matrices $A_k, B_k, C_k$ for the nominal values of $\hat{x}_{t+k}, \hat{u}_{t+k}$ along the prediction horizon. The matrices $A_k$ and $B_k$ are

generated as outputs from the function $f$, which is supplied with a specific input sequence. Additionally, the matrix $C_k$ is produced as an output from the function $g$ when it is given $x_{j+1}$ as an input. The optimization algorithm then solves the quadratic programming problem to determine the optimal sequence $\{y_t^*, ..., u_{t+n_m-1}^*\}$. The problem is formulated as:

$$\min \sum_{k=0}^{n_p-1} ||W_y(y_{t+k+1} - r_{t+k+1})||_2^2 + ||W_{\Delta u}\Delta u_{t+k}||_2^2 +$$
$$||W_u(u_{t+k} - u_{t+k}^r)||_2^2$$
$$\text{s.t. } x_{j+1} = A_k[x_j' 1]' + Bku_j$$
$$y_{j+1} = Ck[x_{j+1}' 1]'$$
$$\Delta u_j = u_j - u_{j-1}, \quad j = t+k, \quad k = 0, ..., n_p - 1$$
$$Linear \quad constraint \quad on \quad \Delta u_{t+k}$$
$$\Delta u_{t+k} = 0, n_m \leq k < n_p$$
$$(3)$$

After solving the optimization problem, the current input $u_t$ is set to the optimal input $u_t^*$ and the nominal input sequence is updated as $\hat{u}_{t+k} = u_{t+k}^*$ for $1 \leq k \leq n_m - 1$ and $\hat{u}_{t+k} = u_{t+n_m-1}^*$ for $n_m \leq k \leq n_p - 1$. The output of the MPC algorithm is the command input $u_t$, along with the updated nominal input sequence $\{\hat{u}_{t+1}, ..., \hat{u}_{t+n_p}\}$. In this study, we employed the Limited-Memory Broyden–Fletcher–Goldfarb–Shanno (L-BFGS-B) algorithm as our optimization technique, which is available in the *scipy* library in Python [16].

To ensure the safety of our approach, we defined specific ranges for the infusion rates and the desired MAP target within the control design. These constraints were incorporated into the optimization process, ensuring that the control actions remain within safe and clinically relevant parameters. This reduces the risk of adverse effects and enhances the clinical applicability of our method. Providing a flexible and robust approach for controlling multivariable systems under constraints on process variables, the detailed MPC algorithm is particularly effective for managing nonlinear systems.

## III. RESULTS

The proposed automated fluid management system is a VAE-based MPC that encompasses the RNSSM model and the MPC controller. The RNSSM model, using fluid and hemorrhage rates as inputs and MAP response as the output, constructs robust nonlinear state-space models that are highly conducive to closed-loop control design. Incorporating the RNSSM model, the MPC controller determines the optimal fluid infusion dosages in different hemorrhage scenarios.

The dataset used for developing the RNSSM models was sourced from an animal study approved by the Institutional Animal Care and Use Committee at the University of Texas Medical Branch [17]. In this study, different sheep underwent high and medium hemorrhage procedures accompanied by fluid infusion. The study involved sheep subjected to varying degrees of hemorrhage and fluid infusion. The MAP was recorded every five minutes throughout the 180-minute study duration. Initial hemorrhage rates were set at 25 ml/kg for

the first 15 minutes, representing a severe hemorrhage event. Hemorrhage was then halted, except for two instances at 50 and 70 minutes into the study, where a rate of 5 ml/kg was applied to each subject for five minutes. These instances represented potential medium hemorrhage events during patient transportation. Therefore, a total of 35 ml/kg of blood was hemorrhaged from each subject during the aforementioned three time periods. Fluid resuscitation commenced 30 minutes into the study using lactated Ringer's solution, a type of crystalloid fluid, and continued until the study's conclusion.

The RNSSM model was fine-tuned and tested on three subjects from the animal study. The training dataset for the RNSSM consisted of 1,800 samples, with an early-stopping strategy employed using 10% of the training dataset to verify the stopping criterion.

The RNSSM model was designed to replicate the MAP in response to infusion and hemorrhage. Therefore, we used infusion and hemorrhage as inputs to the model, and MAP as both the input and target for the model. During the testing process, we used the same infusion and hemorrhage schedule as input to the bridge network, i.e., estimated function $f$ in RNSSM. We then estimated MAP using the estimated states from $f$ and the system inputs, i.e., infusion and hemorrhage.

The MAP responses from the RNSSM model, along with the actual MAP measured from the animal study, are demonstrated in Fig. 2 for the three sample subjects.

Following the design of the RNSSM model, the MPC controller was subsequently developed for each subject. The cost function parameters, $W_y$, $W_{\Delta u}$, and $W_u$, were set to 1, 0.05, and 0.001, respectively. The control and prediction horizons were set to be 5 and 10, respectively. The desired MAP value was set to 80 mmHg for each subject. The MPC controller was designed to minimize the cost function of (2) by taking into account the problem constraints as shown in (2).

For the comparison study, we designed a proportional-integral-derivative (PID) controller for the same subjects. The proportional (P) component provides an immediate response to any changes in the error between the actual and desired MAP; the integral (I) component focuses on the MAP steady-state error, enhancing the controller's ability to reach and maintain the desired set point over time; the derivative (D) component anticipates future MAP error by evaluating the rate of the error, which aids in reducing overshoot and enhancing the system's stability. The PID was designed to reach the same desired MAP as the MPC controller. The PID gains were carefully tuned to achieve the best performance for each subject. The MAP responses and fluid infusion dosages from the MPC and PID for each subject are shown in Figs.3 and 4, respectively.

TABLE I
PERFORMANCE METRICS OF THE RNSSM MODEL FOR EACH SUBJECT

| | **MAE** (%) | **MDAPE** (%) | **RMSE** (%) |
|---|---|---|---|
| Subject 1 | 9.15 | 4.56 | 12.13 |
| Subject 2 | 6.29 | 4.25 | 9.08 |
| Subject 3 | 11.20 | 7.58 | 12.11 |

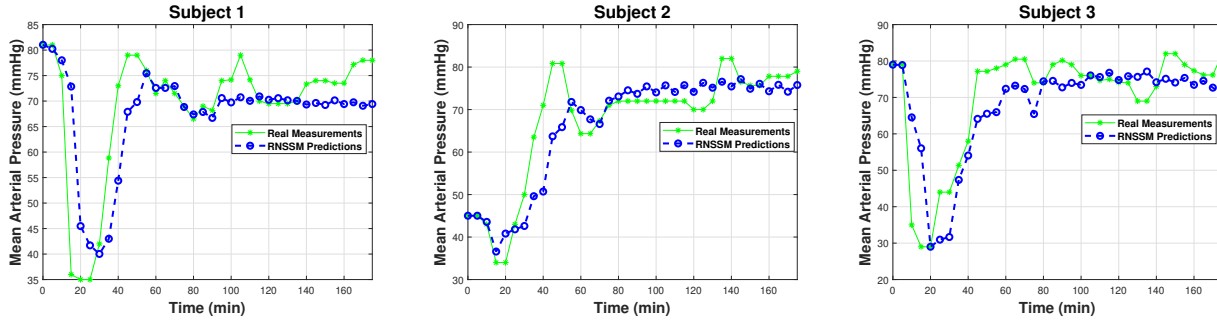

Fig. 2. Mean arterial pressure (MAP) responses predicted by the RNSSM model along with the real MAP measurements

TABLE II
PERFORMANCE METRICS OF THE MPC AND PID CONTROLLERS FOR
EACH SUBJECT

| | MAE (%) | | MDAPE (%) | | RMSE (%) | |
|---|---|---|---|---|---|---|
| | MPC | PID | MPC | PID | MPC | PID |
| Subject 1 | 7.03 | 19.64 | 1.53 | 15.50 | 11.46 | 22.21 |
| Subject 2 | 3.68 | 8.56 | 0.58 | 6.68 | 7.97 | 11.25 |
| Subject 3 | 4.25 | 14.81 | 2.19 | 12.53 | 7.54 | 15.87 |

## IV. DISCUSSION

The close alignment between the MAP predicted by the RNSSM model and the measured MAP, as shown in Fig. 2, confirms the model's ability to accurately trace real-time series data trend and effectively capture the MAP variations induced by hemorrhage. This highlights the model's robustness in the face of external perturbations. The error between the model's predictions and the actual measurements may be attributed to inherent uncertainties and variability in the physiological system. However, the overall trend of hemodynamic response is well captured by the RNSSM, demonstrating its predictive capabilities. To quantitatively assess the model's precision, three performance metrics, including root mean square error (RMSE), mean absolute error (MAE), and median absolute percentage error (MDAPE), were selected. The results, enumerated in Table II, indicate the RNSSM model's capability to robustly and accurately encapsulate MAP responses throughout hemorrhage resuscitation.

Comparing the performance of MPC and PID controllers, as shown in Figs. 3 and 4, reveals that while both controllers showed promising results in controlling the MAP response, the MPC performed superiorly. For all three subject, the MPC controller achieved a MAP response closer to the target level of 80 mmHg, underscoring its higher accuracy. Moreover, the MPC controller maintained more stable MAP values and infusion rates over time compared to the PID controller.

The RMSE, MAE, and MDAPE for both PID and MPC controllers are shown in Table III. The results clearly demonstrate that the MPC controller outperforms the PID in terms of these performance metrics. This can be attributed to the optimization method and predictive capability of the MPC, features that are absent in PID, allowing the MPC to anticipate future MAP values and adjust dosages accordingly. Therefore, the

MPC controller provides more accurate and reliable control of hemodynamic responses during hemorrhagic events, compared to the PID controller.

The results of our model and controller imply that a VAE-based MPC is a powerful tool for understanding and predicting hemodynamic responses in hypovolemic scenarios, showing great promise for further extension and future applications in healthcare automation. However, there are limitations that need to be considered. The study was conducted on a small sample size of three subjects, which may not fully capture the variability in the larger population. Additionally, the performance of the MPC, liked other model-based controllers, heavily relies on the accuracy of the model used. If the model does not properly represent the system, the performance of the MPC would be degraded. Furthermore, this work only used MAP response as the hemodynamic endpoint. Including variables such as heart rate or blood volume can potentially enhance hemodynamic predictions.

In response to these challenges, our immediate future work focuses on validating the VAE-based MPC with a larger hemorrhage resuscitation dataset, which could enhance the generalizability of our approach. Additionally, We plan to incorporate other hemodynamic variables into our models and controllers to better capture physiological variability and enhance the accuracy of hemodynamic predictions. The choice of MPC is particularly advantageous due to its ability to handle multiple-input/multiple-output systems, making it a promising tool for future advancements. Furthermore, we will test our algorithm in the fluid resuscitation hardware-in-the-loop test bed that we developed and used in our previous work [2]. This test bed provides a more realistic testing environment to further assess the safety and efficacy of our proposed algorithm before clinical application.

## V. CONCLUSION

We presented a pioneering approach to automated fluid management using an RNSSM model and an MPC controller. The RNSSM framework successfully models hemodynamic responses during hemorrhage resuscitation, while the MPC controller leverages its predictive capabilities for precise control of fluid dosages. The proposed method captures complex, nonlinear relationships within physiological systems, allowing

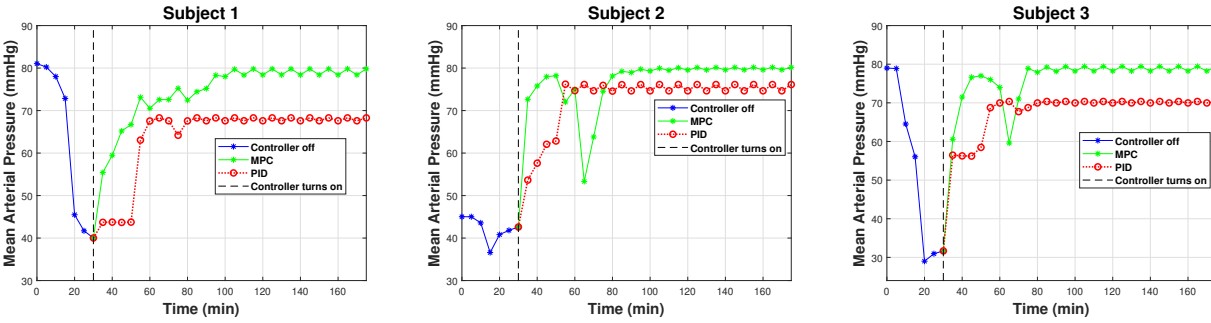

Fig. 3. Mean arterial pressure (MAP) responses from the MPC and PID controllers

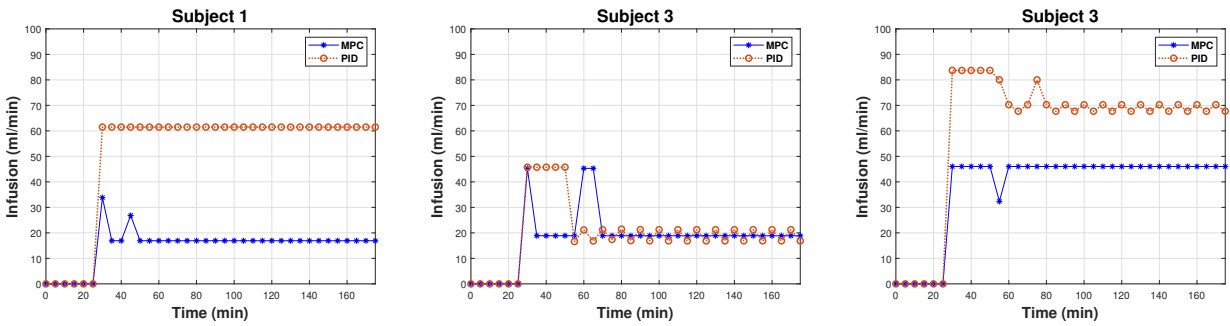

Fig. 4. Fluid infusion dosages recommended by the MPC and PID controllers

for more precise and anticipatory control of fluid delivery. The VAE-based MPC is data-efficient, requiring less data to build an effective model compared to model-free methods, while offering greater stability and robustness than low-order lumped-parameter models. Simulations results demonstrated the potential of this approach in improving the safety and efficacy of fluid resuscitation in critical care. Future work will focus on fine-tuning the model parameters, incorporating further hemodynamic variables, and assessing the VAE-based MPC on a larger clinical dataset.

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
