# OpenReview forum: "Variational Autoencoder-Based Model Predictive Control for Automated Fluid Resuscitation"
_IEEE.org/EMBS/BHI/2024/Conference — IEEE BHI'24_

### Official Review · Reviewer_st1L · 2024-08-01
**Variational Autoencoder-Based Model Predictive Control for Automated Fluid Resuscitation**

**Overall Rating:** 7
**Confidence:** 5

**Other Quality Metrics:**

All 4 items are excellent

**Questions For The Authors:**

You have a great paper.

**Strengths:**

The use of machine learning in this problem is well addressed.

**Summary Of The Paper:**

The paper presents limited novelty.

**Weaknesses:**

It is not clear what are the alternative methods to solve this.

---

### Official Review · Reviewer_zKmc · 2024-08-10
**The title is reasonable.**

**Overall Rating:** 6
**Confidence:** 4

**Other Quality Metrics:**

Good.

**Questions For The Authors:**

It needs to be proved with more evidence for the safety of the real application for the clinic.

**Strengths:**

It demonstrated the potential of the approach in improving the safety and efficacy of fluid resuscitation in critical care settings.

It looks great reseach that can improve the safety in clinical system.

**Summary Of The Paper:**

The paper described a pioneering approach to automated fluid management using an RNSSM model and an MPC controller. The RNSSM framework successfully models hemodynamic responses during hemorrhage resuscitation, while the MPC controller leverages its predictive capabilities for precise control of fluid dosages. The proposed method can capture complex, nonlinear relationships within physiological systems, allowing for better precise and anticipatory control of fluid delivery. The result showed that the VAE-based MPC is data-efficient, requiring less data to build an effective model compared to model-free methods, while offering greater stability and robustness than low-order lumped-parameter models. Simulations results demonstrated the potential of this approach in improving the safety and efficacy of fluid resuscitation in critical care.

**Weaknesses:**

The lower number of the subjects was applied. It should be tested with more subjects.

 It still needs to check with larger clinical dataset.

---

### Official Review · Reviewer_Pcw8 · 2024-08-27
**Good performance with complex model, but there could be further understanding of each part**

**Overall Rating:** 6
**Confidence:** 4

**Other Quality Metrics:**

Clarity of writing:

Great

Clinical Significance:

Good

Methodological Novelty:

Good

Experiments and Results:

Fair

**Questions For The Authors:**

Since the encoder and decoder and bridge has relatively shallow layers and a moderate number of nodes among the subjects and it is mentioned in the paper that good modeling is critical for MPC performance, some gap seems to be remained on why latent representation via VAE is necessary instead of learning a transition directly with neural network. Some ablation studies or concerns either on data size requirements or something else should be addressed. Besides, in the experiments of subject 2, based on the performance the MPC seems to use around 72mmHg as its target arterial pressure, though the true target should be 80mmHg. Is there some intuition on why that is happening? For the baseline comparison, many planning algorithms exist. Is there some specific reason why PID is the only selected baseline or why it is sufficient to be chosen as the only baseline?

**Strengths:**

It is a complex model with good theoretical background that integrates multiple components and seems effective based on the experiments.

**Summary Of The Paper:**

The paper proposes a method that learns a latent space representation of the hemodynamics via variational autoencoder, uses Quasi-Linear Parameter-Varying structure to model the latent space transition, and deploys model predictive control to achieve automated fluid resuscitation.

**Weaknesses:**

However, it is unclear how the good performance is related to each component.

---

### Decision · Program_Chairs · 2024-09-23

Accept